# Prevalence and Correlates of Behavioral Non-Communicable Diseases Risk Factors among Adolescents in the Seychelles: Results of a National School Survey in 2015

**DOI:** 10.3390/ijerph16152651

**Published:** 2019-07-25

**Authors:** Supa Pengpid, Karl Peltzer

**Affiliations:** 1ASEAN Institute for Health Development, Mahidol University, Nakhon Pathom 73170, Thailand; 2Deputy Vice Chancellor Research and Innovation Office, North West University, Potchefstroom 2531, South Africa

**Keywords:** behavioral risk factors, non-communicable diseases, adolescents, Seychelles

## Abstract

The aim of this study was to investigate the prevalence and correlates of behavioral non-communicable disease (NCD) risk factors among a national sample of school-going adolescents in the Seychelles. Cross-sectional data were analyzed from 2540 school adolescents (median age 14 years, interquartile range = 2), in the Seychelles “Global School-Based Student Health Survey (GSHS)” in 2015. Behavioral NCD risk factors (current tobacco use, current alcohol use, inadequate fruit and vegetable consumption, soft drink consumption, overweight or obesity, physical inactivity, and leisure-time sedentary behavior) were assessed by self-report. Among the seven individual behavioral risk factors, the highest prevalence was physical inactivity (82.7%), followed by daily soft drink consumption (68.3%), inadequate fruit and vegetable consumption (60.9%), leisure-time sedentary behavior (51.0%), current alcohol use (47.6%), overweight or obesity (28.2%), and current tobacco use (23.4%). The total mean number of behavioral NCD risk factors was 3.6 (Standard Deviation = 1.3), and the proportion of co-occurrence of having three or more behavioral NCD risk factors was 80.7%. In adjusted linear regression analysis, male sex, older age, and psychological distress were positively, and school attendance and peer support were negatively associated with the total number of behavioral NCD risk factors. A high prevalence of multiple behavioral NCD risk factors were found and several associated factors were identified, such as male sex, older age, psychological distress, school truancy, and lack of peer support, which may help in aiding intervention programs in this population.

## 1. Introduction

“Non-communicable diseases (NCDs), such as cardiovascular diseases, cancers, respiratory diseases, and diabetes, cause 71% of all deaths globally, and over 85% in low- and middle-income countries” [1]. Behavioral risk factors such as “tobacco use, physical inactivity, the harmful use of alcohol and unhealthy diets all increase the risk of dying from a NCD” [1]. In the Seychelles, a small country in the Indian Ocean (total population 95,000), NCDs are estimated to account for 81% of all deaths [2]. In the 2007 Global School-Based Student Health Survey (GSHS) in the Seychelles, 22% and 11% of boys and girls, respectively were current smokers and 60% and 56% of boys and girls, respectively had consumed alcohol in the past few month [3]. There has been a large increase in overweight or obesity among adolescents in the Seychelles to 20.0% in boys and 23.6% in girls in 2016 [4]. Stringhini et al. [5], found in a sample of adults (25–45 years) in the Seychelles that 20.8% were smokers, 40.9% were physically active, and 25.1% were obese. In a study among adolescents in 65 low-income and middle-income countries (between 2003 and 2011), the prevalence of behavioral NCD risk factors was 12.1% for tobacco use, 15.7% for alcohol use, 74.3% for low fruit and vegetable intake, 71.4% for low physical activity, and 7.1% for obesity [6]. The pooled regional prevalence of exposure to three or more behavioral NCD risk factors was 6.1% in the Africa region, and was the highest in the Seychelles in 2007 (19.6%) [6]. In a study among adolescents in Nepal, the prevalence of current alcohol use was 5.3%, current smoking was 6.0%, insufficient fruit and vegetable intake was 95.3%, overweight or obesity was 6.7%, and inadequate physical activity was 84.8%; and 11.2% had three or more risk factors [7]. However, no information is available on the prevalence and clustering of seven behavioral NCD risk factors among adolescents in the Seychelles. Knowledge and understanding of the prevalence and clustering of seven behavioral NCD risk factors may have relevant policy implications in identifying and targeting particular risk groups in order to prevent NCDs in this adolescent population.

Factors associated with a higher likelihood of co-occurrence of behavioral NCD risk factors among adolescents include the older age group [8,9,10], males [7], low socioeconomic level [9], and not attending physical education classes [10,11]. Factors associated with individual risk behaviors for current tobacco use include males [7], older adolescents [10], and not being active during a physical education class [10]; and for current alcohol use include males [7], older adolescents [10], and higher economic status [10]. Factors associated with inadequate fruit and vegetable consumption may include non-participation in organized physical activity [10], insufficient physical activity [7], and overweight/obesity [7].

Psychological distress has been associated with various behavioral NCD risk factors, e.g., tobacco use [12], alcohol use [13], inadequate fruit and vegetable consumption [14], soft drink consumption [15], overweight or obesity [16], physical inactivity [14], sedentary behavior [14], and multiple health risk behavior [14]. Protective factors for individual and co-occurring behavioral NCD risk factors may include school attendance [17], attending physical education classes, and peer and parental support. For example, attending physical education classes was associated with adhering to physical activity recommendations among adolescents in Brazil [11,18], and with recommendations of fruit consumption [11]. Parental support was found among adolescents to be protective from substance use [19], inadequate fruit and vegetable consumption, and physical inactivity [20]. The aim of this study was to investigate the prevalence and correlates of behavioral NCD risk factors in a national sample of school-going adolescents in the Seychelles.

## 2. Methods

### 2.1. Sample and Procedure

The study included a secondary analysis of the 2015 Seychelles GSHS, which employed “a two-stage cluster sample design to produce data representative of all students in grades S1–S5 in Seychelles.” [21]. “At the first stage, schools were selected with probability proportional to enrolment size. At the second stage, classes were randomly selected and all students in selected classes were eligible to participate, regardless of their age, and completed a self-administered questionnaire in their language under the supervision of trained external survey administrators.” [21] The study protocol was approved by the Ministry of Health, Seychelles, and a national ethics committee, and “necessary approvals and permission, including informed consent, were obtained from the participating schools, parents and students before the survey was administered.” [21]. 

### 2.2. Measures

The GSHS measure was used in this survey, as described in Appendix A
Table A1 [22]. Inadequate fruit and vegetable consumption was classified as less than five or more servings in a day [23]. Overweight or obesity was defined as “more than +1 standard deviation (SD) from the median body mass index by age and sex” [24]. Inadequate physical activity was defined as not daily “at least 60 min of moderate to vigorous-intensity physical activity” [25]. “Sedentary behaviour was defined as spending 3 or more hours per day sitting” [26]. The psychological distress items (no close friends, loneliness, anxiety, suicidal ideation, and suicide attempt) were summed, and grouped into 0 = 0, 1 = 1 single and 2–5 = 2 multiple. The four items on parental or guardian support were summed, and classified into three groups, 0–1 low, 2 medium, and 3–4 high support. 

### 2.3. Data Analysis

Data analysis was conducted with STATA software version 15.0 (Stata Corporation, College Station, TX, USA), taking into account the complex study design. Multivariable Poisson regression and linear regression were used to assess the associations between sociodemographic, psychological distress, protective factors, and individual and total number of behavioral risk factors. Missing cases were excluded from the analysis. *p*-values <0.05 were considered significant. 

## 3. Results

### 3.1. Sample Characteristics and Distribution of Behavioral Risk Factors

The sample included 2540 school-going adolescents (median age 14 years, interquartile range = 2); the overall response rate was 82% [21]. About half of the participants (52.7%) were females and 47.3% were male, 28.5% had sometimes, mostly, or always experienced hunger or food insecurity (or a proxy for low economic status), 22.4% had one and 18.5% multiple psychological distress items. Protective factors included: 69.8% had attended school in the past month, 40.1% had attended at least two physical education classes in the past week, 22.9% received peer support, and 54.2% had two or more parental support indicators. 

Among the seven individual behavioral risk factors, the highest prevalence was physical inactivity (82.7%), followed by daily soft drink consumption (68.3%), inadequate fruit and vegetable consumption (60.9%), leisure-time sedentary behavior (51.0%), current alcohol use (47.6%), overweight or obesity (28.2%), and current tobacco use (23.4%). The total mean number of behavioral NCD risk factors was 3.6 (SD = 1.3), and the proportion of co-occurrence the behavioral NCD risk factors was 0 = 0.8%, 1 = 4.3%, 2 = 14.2%, 3 = 28.1%, 4 = 26.2%, 5 = 19.4%, 6 = 6.2%, and 7 = 0.8%; and 80.7% had three or more behavioral NCD risk factors (see Table 1).

### 3.2. Associations with Tobacco Use, Alcohol Use, Fruit and Vegetable Consumption, and Soft Drink Consumption

In adjusted Poisson regression analysis, male sex, older age, psychological distress, and attending physical education classes were positively associated, and school attendance and parental support negatively associated with current tobacco use and/or current alcohol use. Older age and rarely experience hunger were positively associated, and one psychological distress item, attending physical education classes, and parental support were negatively associated with inadequate fruit and vegetable consumption. School attendance was protective from soft drink consumption (see Table 2).

### 3.3. Associations with Overweight or Obesity, Physical Inactivity, Leisure-Time Sedentary Behavior, and Total Number of Behavioral NCD Risk Factors

In adjusted Poisson regression analysis, being 16 years and older decreased the odds for being overweight or obese. Male sex, attending physical education classes, and peer and parental support were negatively associated with inadequate physical activity. Female sex, older age, and peer support increased the odds for leisure-time sedentary behavior. Male sex, older age, and psychological distress items were positively associated, and school attendance and peer support were negatively associated with the total number of behavioral NCD risk factors (see Table 3)

## 4. Discussion

The study found among school-going adolescents in the Seychelles a high co-occurrence of NCD behavioral risk factors of a mean of 3.6 (range 0–7), and 80.7% had three to seven risk factors, which is higher than the pooled prevalence of exposure to three to five NCD risk factors in the Africa region (6.1%), in the Seychelles in 2007 (19.6%) [6], and in Nepal (11.2%) [7]. Among the seven individual behavioral NCD risk factors measured among adolescents, the prevalence of physical inactivity (82.7%) was higher than in the older multi-country study (71.4%) [6], inadequate fruit and vegetable consumption was lower (60.9% vs. 74.3%) [6], and current tobacco use (23.4% vs. 12.1%) and alcohol use was higher than in the multi-country study (47.6% vs. 15.7%) [6]. The prevalence of overweight or obesity (28.2%) was higher than in a previous study in the Seychelles (21.8%) [4], and in Nepal (6.7%) [7]. Based on these findings of high co-occurrence and individual behavioral NCD risk factors, we are calling for urgent action for interventions in order to reduce both simultaneous and individual behavioral NCD risk factors in the Seychelles.

Consistent with previous studies [7,8,9,10], this study found that being male and of an older age increased the odds for the co-occurrence of health risk behaviors, and tobacco and alcohol use among adolescents. Female adolescents engaged in more sedentary behavior and physical inactivity than male adolescents. Some of these gender differences may be linked to socio-cultural conditions influencing the identity of males and females, such as a preference of physical activity among males and more sedentary behaviors among females [10]. Older adolescents may have acquired greater autonomy than younger adolescents, increasing their access to tobacco and alcohol use [8]. 

This study found that psychological distress was associated with multiple behavioral NCD risk factors and tobacco and alcohol use. The multiple affective behavior change model [27] may help in understanding the relationship between psychological distress and health risk behaviors among adolescents. “According to this model, a reciprocal relationship exists between mood and the engagement in mood-regulating behaviours (i.e., substance use, high energy snacking, and sedentary behaviour).” [14].

Unlike some previous studies [10,11], this study did not find an association between not having physical education classes and multiple behavioral NCD risk behaviors. In our study, attending physical education classes was positively associated with tobacco use, which contradicts the findings of a previous study [10]. Consistent with previous research [10,11,18], this study found that attending physical education classes was protective from inadequate fruit and vegetable consumption and inadequate physical activity.

School attendance was protective from the total number of behavioral NCD risk factors and three individual risk behaviors (tobacco, alcohol, and soft drink consumption). This finding reinforces the protective environment of schools regarding substance use and soft drink consumption [17]. Seychelles adopted “A National School Nutrition Policy” in 2012 that bans the sale of “sugar-sweetened soft drinks in schools.” [28]. Moreover, this study found that peer support was protective from multiple NCD risk behaviors and inadequate physical activity. Consistent with previous studies [19,20], this study found that parental support was protective from tobacco use, alcohol use, inadequate fruit and vegetable consumption, and inadequate physical activity. This finding supports the idea to promote peer and parental support in order to reduce behavioral NCD risk behaviors.

## 5. Study Limitations

The study used cross-sectional school data, which precludes causal inferences. Moreover, all measures were self-reported, and future studies should include objective measures, such as anthropometry, health exam data, blood pressure, and biochemical tests. 

## 6. Conclusions

A high prevalence of multiple behavioral NCD risk factors were found, as were several associated factors identified, such as male sex, older age, psychological distress items, school truancy, and lack of peer support, which may help in aiding intervention programs in this adolescent population in the Seychelles.

## Figures and Tables

**Table 1 ijerph-16-02651-t001:** Sample characteristics and distribution of behavioral risk factors of non-communicable diseases in Seychelles.

Variable	Sample	Current Tobacco Use	Current Alcohol Use	Inadequate Fruit and Vegetable Consumption	Soft Drink Consumption	Overweight or Obesity	Inadequate Physical Activity	Leisure-Time Sedentary Behavior	Total Behavioral Risk Factors
	*N* (%)	%	%	%		%	%	%	Mean (Standard Deviation)
Sociodemographics									
All	2540	23.4	47.6	60.9	68.3	28.2	82.7	51.0	3.6 (1.3)
Sex									
Female	1337 (52.7)	17.9	48.1	62.3	68.7	29.4	85.4	54.4	3.7 (1.3)
Male	1202 (47.3)	28.9	47.1	59.4	67.8	26.9	79.9	47.5	3.5 (1.3)
Age									
≤13	1148 (41.9)	19.4	40.0	56.5	68.5	30.4	84.1	44.0	3.4 (1.3)
14	473 (19.5)	22.4	45.6	58.3	72.0	28.5	82.1	49.9	3.6 (1.3)
15	482 (20.5)	28.3	55.6	65.9	65.2	28.0	81.1	58.1	3.8 (1.3)
≥16	435 (18.0)	28.1	57.5	68.0	67.3	23.5	82.1	60.4	3.9 (1.3)
Experience hunger									
Never	1406 (56.1)	23.2	45.6	60.9	68.3	25.6	81.7	51.1	3.6 (1.3)
Rarely	396 (15.4)	21.5	47.9	69.9	69.6	30.8	83.1	49.5	3.8 (1.3)
Sometimes-always	708 (28.5)	24.2	50.6	55.8	67.7	31.2	84.2	52.2	3.6 (1.3)
Psychological distress									
0	1376 (59.2)	15.9	42.6	63.4	66.7	28.2	80.6	48.4	3.5 (1.3)
1	511 (22.4)	25.3	52.6	56.2	70.0	28.1	82.3	54.2	3.8 (1.3)
2–5	430 (18.5)	33.2	55.4	58.6	70.1	27.7	85.8	55.4	3.8 (1.4)
Protective factors									
School attendance	1664 (69.8)	15.1	40.8	61.9	65.2	29.3	81.7	48.9	3.5 (1.3)
Attending physical education classes/week									
0	438 (18.7)	28.7	48.3	60.3	71.6	29.2	90.0	49.7	3.7 (1.3)
1	1001 (41.3)	16.9	44.9	64.8	65.8	27.4	83.9	50.2	3.6 (1.3)
≥2	985 (40.1)	26.4	50.0	56.0	69.0	28.9	77.8	52.4	3.6 (1.4)
Peer support	566 (22.9)	22.6	47.3	57.8	67.7	28.8	77.0	54.6	3.6 (1.4)
Parental support									
0–1	1024 (45.8)	30.7	42.6	65.0	69.8	27.6	86.5	51.2	3.8 (1.3)
2	612 (27.1)	17.8	52.6	58.2	66.3	29.9	79.4	53.5	3.6 (1.4)
3–4	630 (27.1)	11.4	55.4	56.3	67.0	28.4	78.9	46.2	3.3 (1.3)

**Table 2 ijerph-16-02651-t002:** Associations with behavioral non-communicable disease (NCD) risk factors (tobacco, alcohol, fruit and vegetable consumption, and soft drink consumption).

Variable	Current Tobacco Use	Current Alcohol Use	Inadequate Fruit and Vegetable Consumption	Soft Drink Consumption
	APR (95% CI)	APR (95% CI)	APR (95% CI)	APR (95% CI)
Sociodemographics				
Sex				
Female	1 (Reference)	1 (Reference)	1 (Reference)	1 (Reference)
Male	1.57 (1.32, 1.86) ***	0.97 (0.87, 1.02)	0.95 (0.88, 1.03)	0.98 (0.91, 1.05)
Age				
≤13	1 (Reference)	1 (Reference)	1 (Reference)	1 (Reference)
14	1.28 (0.92, 1.77)	1.18 (1.01, 1.39) *	1.03 (0.91, 1.17)	1.05 (0.94, 1.18)
15	1.50 (1.19, 1.88) ***	1.41 (1.18, 1.67) ***	1.13 (0.98, 1.31)	0.95 (0.84, 1.07)
≥16	1.58 (1.18, 2.12) **	1.48 (1.28, 1.71) ***	1.17 (1.06, 1.29) **	0.97 (0.88, 1.08)
Experience hunger				
Never	1 (Reference)	1 (Reference)	1 (Reference)	1 (Reference)
Rarely	0.89 (0.70, 1.14)	1.05 (0.93, 1.19)	1.14 (1.04, 1.24) **	1.02 (0.94, 1.10)
Sometimes-always	0.77 (0.62 (0.96) *	1.03 (0.91, 1.15)	0.90 (0.80, 1.01)	0.97 (0.90, 1.05)
Psychological distress				
0	1 (Reference)	1 (Reference)	1 (Reference)	1 (Reference)
1	1.47 (1.21 (1.78) ***	1.14 (1.00, 1.30) *	0.90 (0.83, 0.98) *	1.04 (0.96, 1.12)
2–5	1.98 (1.58, 2.48) ***	1.20 (1.04, 1.38) *	0.94 (0.85, 1.03)	1.03 (0.94, 1.13)
Protective factors				
School attendance	0.51 (0.43, 0.61) ***	0.70 (0.66, 0.78) ***	1.08 (0.99, 1.17)	0.90 (0.84, 0.96) **
Attending physical education classes/≥2 week	1.24 (1.07, 1.44) **	1.08 (0.98, 1.19)	0.91 (0.84, 0.99) *	1.01 (0.94, 1.09)
Peer support	1.12 (0.90, 1.40)	1.02 (0.91, 1.15)	0.93 (0.84, 1.02)	0.92 (0.85, 1.01)
Parental support				
0–1	1 (Reference)	1 (Reference)	1 (Reference)	1 (Reference)
2	0.64 (0.50, 0.82) ***	0.85 (0.75, 0.95) **	0.92 (0.83, 1.01)	0.94 (0.87, 1.01)
3–4	0.45 (0.32, 0.61) ***	0.74 (0.64, 0.86) ***	0.88 (0.80, 0.97) **	0.96 (0.89, 1.04)

APR = adjusted prevalence ratio; CI = confidence interval; *** *p* < 0.001, ** *p* < 0.01, * *p* < 0.05.

**Table 3 ijerph-16-02651-t003:** Associations with behavioral NCD risk factors (overweight or obesity, physical inactivity, leisure-time sedentary behavior, and total number of behavioral NCD risk factors).

Variable	Overweight or Obesity	Inadequate Physical Activity	Leisure-Time Sedentary Behavior	Total Behavioral Risk Factors
	APR (95% CI)	APR (95% CI)	APR (95% CI)	Adjusted Beta (95% CI)
Sociodemographics				
Sex				
Female	1 (Reference)	1 (Reference)	1 (Reference)	Reference
Male	0.97 (0.83, 1.13)	0.92 (0.87, 0.97) ***	0.84 (0.77, 0.93) ***	0.16 (0.07, 0.26) ***
Age				
≤13	1 (Reference)	1 (Reference)	1 (Reference)	Reference
14	0.98 (0.81, 1.18)	0.98 (0.92, 1.04)	1.11 (0.94, 1.31)	0.08 (−0.04, 0.21)
15	0.88 (0.71, 1.09)	0.95 (0.89, 1.01)	1.35 (1.18, 1.55) ***	0.14 (0.01, 0.27) *
≥16	0.81 (0.67, 0.98) *	0.98 (0.92, 1.05)	1.47 (1.26, 1.71) ***	0.21 (0.29, 0.53) ***
Experience hunger				
Never	1 (Reference)	1 (Reference)	1 (Reference)	Reference
Rarely	1.18 (0.97, 1.44)	1.02 (0.96, 1.08)	0.93 (0.83, 1.06)	0.10 (−0.01, 0.22)
Sometimes-always	1.16 (0.99, 1.36)	1.01 (0.96, 1.06)	1.00 (0.90, 1.11)	−0.03 (−0.15, 0.08)
Psychological distress				
0	1 (Reference)	1 (Reference)	1 (Reference)	Reference
1	1.01 (0.83, 1.23)	1.01 (0.96, 1.06)	1.09 (0.97, 1.22)	0.18 (0.05, 0.30) **
2–5	1.04 (0.87, 1.23)	1.03 (0.98, 1.09)	1.06 (0.94, 1.19)	0.58 (0.44, 0.72) ***
Protective factors				
School attendance	1.10 (0.92, 1.32)	0.98 (0.93, 1.03)	0.91 (0.83, 1.00)	−0.42 (−0.57, −0.27) ***
Attending physical education classes/≥2 week	1.04 (0.87, 1.23)	0.90 (0.86, 0.94) ***	1.04 (0.95, 1.14)	−0.06 (−0.17, 0.04)
Peer support	1.02 (0.88, 1.20)	0.94 (0.90, 0.99) *	1.11 (1.01, 1.23) *	−0.20 (−0.32, −0.08) ***
Parental support				
0–1	1 (Reference)	1 (Reference)	1 (Reference)	Reference
2	1.11 (0.93, 1.34)	0.94 (0.89, 0.99) *	1.02 (0.93, 1.14)	−0.08 (−0.20, 0.05)
3–4	1.04 (0.85, 1.26)	0.93 (0.88, 0.99) *	0.90 (0.81, 1.01)	−0.16 (−0.33, 0.01)

APR = adjusted prevalence ratio; CI = confidence interval; *** *p* < 0.001, ** *p* < 0.01, * *p* < 0.05.

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
