# Peer review of "Prevalence and Correlates of Behavioral Non-Communicable Diseases Risk Factors among Adolescents in the Seychelles: Results of a National School Survey in 2015"

_ijerph, 2019, doi:10.3390/ijerph16152651_

Round 1

Reviewer 1 Report

The introduction does not pint out what is novel about this manuscript.  Seems like there has been considerable previous research done in this area and it is not clear what new things the current project addresses.  the intro should work toward identification of a gap in the literature that is important to fill and that this ms can fill.

The sampling and survey construction sections need expansion.  This should include resonse rate

Table 1 could go in supplemental materials section

Author Response

Reviewer I

Comments and Suggestions for Authors

The introduction does not pint out what is novel about this manuscript.  Seems like there has been considerable previous research done in this area and it is not clear what new things the current project addresses.  the intro should work toward identification of a gap in the literature that is important to fill and that this ms can fill.

Response: below is added

However, no information is available on the prevalence and clustering of seven behavioural NCD risk factors among adolescents in the Seychelles. Knowledge and understanding of the prevalence and clustering of seven behavioural NCD risk factors may have relevant policy implications in identifying and targeting particular risk groups in order to prevent NCDs in this adolescent population.

The sampling and survey construction sections need expansion.  

Response: more is added

This should include resonse rate

Response: this is added in the results section

Table 1 could go in supplemental materials section

Response: changed accordingly

Reviewer 2 Report

The cross-sectional study were addressed from 2540 school adolescents (median age 14 years, IQR=2), using data from Global School-Based Student Health Survey (GSHS) in 2015, Seychelles. The survey conducted by interviewed questionnaires, including 7 behaviours related to non-communicable diseases (NCD): inadequate physical inactivity, daily soft drink consumption, inadequate fruit and vegetable consumption, leisure-time sedentary behaviour, alcohol use, overweight or obesity, and tobacco use. In addition, the survey also included basic demographic data, psychological distress, and protective factors. However, there were no chronic diseases, nor health exam data, such as biochemical tests.

The main problem of this research was missed use of “behavioural non-communicable diseases”. The dependent variables (outcomes) in this study were behavioural risk factors for non-communicable diseases: inadequate physical inactivity, daily soft drink consumption, inadequate fruit and vegetable consumption, leisure-time sedentary behaviour, alcohol use, overweight or obesity, and tobacco use.

I suggest that the authors should re-write the manuscript, and have to change the title, study goal, methods, results, and discussion, regarding to the definition of non-communicable diseases on the website of WHO, as “Noncommunicable diseases (NCDs), also known as chronic diseases, tend to be of long duration and are the result of a combination of genetic, physiological, environmental and behaviours factors. The main types of NCDs are cardiovascular diseases (like heart attacks and stroke), cancers, chronic respiratory diseases (such as chronic obstructive pulmonary disease and asthma) and diabetes.” Thus, I do not find any diseases, or well defined “behavioural NCD” in the article.

Author Response

Reviewer II

Comments and Suggestions for Authors

The cross-sectional study were addressed from 2540 school adolescents (median age 14 years, IQR=2), using data from Global School-Based Student Health Survey (GSHS) in 2015, Seychelles. The survey conducted by interviewed questionnaires, including 7 behaviours related to non-communicable diseases (NCD): inadequate physical inactivity, daily soft drink consumption, inadequate fruit and vegetable consumption, leisure-time sedentary behaviour, alcohol use, overweight or obesity, and tobacco use. In addition, the survey also included basic demographic data, psychological distress, and protective factors. However, there were no chronic diseases, nor health exam data, such as biochemical tests.

Response: The latter is included in the study limitations

The main problem of this research was missed use of “behavioural non-communicable diseases”. The dependent variables (outcomes) in this study were behavioural risk factors for non-communicable diseases: inadequate physical inactivity, daily soft drink consumption, inadequate fruit and vegetable consumption, leisure-time sedentary behaviour, alcohol use, overweight or obesity, and tobacco use.

I suggest that the authors should re-write the manuscript, and have to change the title, study goal, methods, results, and discussion, regarding to the definition of non-communicable diseases on the website of WHO, as “Noncommunicable diseases (NCDs), also known as chronic diseases, tend to be of long duration and are the result of a combination of genetic, physiological, environmental and behaviours factors. The main types of NCDs are cardiovascular diseases (like heart attacks and stroke), cancers, chronic respiratory diseases (such as chronic obstructive pulmonary disease and asthma) and diabetes.” Thus, I do not find any diseases, or well defined “behavioural NCD” in the article.

Response: below is already from WHO stated in the introduction, but behavioural risk factors are now emphasized

“Non-communicable diseases (NCDs), such as cardiovascular diseases, cancers, respiratory diseases, and diabetes, cause 71% of all deaths globally, and over 85% in low- and middle-income countries” [1]. Behavioural risk factors, such as “tobacco use, physical inactivity, the harmful use of alcohol and unhealthy diets all increase the risk of dying from a NCD” [1].

Further, the aim is to look at “behavioural NCD risk factors” such as tobacco use and not medical conditions-risk factors, such as hypertension or dyslipidemia 

Round 2

Reviewer 1 Report

The manuscript is improved with the changes done by the authors   No new problems have emerged. 

Reviewer 2 Report

The authors had used "risk factors" in the title and text. I have no more questions.